Host dietary specialization and neutral assembly shape gut bacterial communities of wild dragonflies

Deb Rittik rittikd@ncbs.res.in debrittik@gmail.com 1
Nair Ashwin 1 2
Agashe Deepa dagashe@ncbs.res.in 1
1 National Centre for Biological Sciences, TIFR , Bangalore , Karnataka , India
2 Shanmugha Arts, Science, Technology & Research Academy (SASTRA University) , Thanjavur , Tamil Nadu , India
Caporaso J Gregory
Electronic publication date: 2019 Nov 18
Publication date: 2019
Volume: 7
Electronic Location ID: e8058
Received 2019 Apr 30; Accepted 2019 Oct 18
Copyright: ©2019 Deb et al.
Copyright year: 2019
Copyright holder: Deb et al.
License: This is an open access article distributed under the terms of the Creative Commons Attribution License, which permits unrestricted use, distribution, reproduction and adaptation in any medium and for any purpose provided that it is properly attributed. For attribution, the original author(s), title, publication source (PeerJ) and either DOI or URL of the article must be cited.
License URL: https://creativecommons.org/licenses/by/4.0/

Keywords: Community assembly, Gut microbiome, Host-microbial interactions, Predator, Specialist, Geographic variation

Funding: National Centre for Biological Sciences Department of Science and Technology, India IFA-13 LSBM-64 Wellcome Trust/DBT India Alliance Fellowship IA/I/17/1/503091 This work was supported by the National Centre for Biological Sciences, the Department of Science and Technology, India (INSPIRE Faculty award IFA-13 LSBM-64 to Deepa Agashe), and a Wellcome Trust/DBT India Alliance Fellowship (grant number IA/I/17/1/503091 to Deepa Agashe). The funders had no role in study design, data collection and analysis, decision to publish, or preparation of the manuscript.

==============================
Host-associated gut microbiota can have significant impacts on host ecology and evolution and are often host-specific. Multiple factors can contribute to such host-specificity: (1) host dietary specialization passively determining microbial colonization, (2) hosts selecting for specific diet-acquired microbiota, or (3) a combination of both. The latter possibilities indicate a functional association and should produce stable microbiota. We tested these alternatives by analyzing the gut bacterial communities of six species of wild adult dragonfly populations collected across several geographic locations. The bacterial community composition was predominantly explained by sampling location, and only secondarily by host identity. To distinguish the role of host dietary specialization and host-imposed selection, we identified prey in the guts of three dragonfly species. Surprisingly, the dragonflies–considered to be generalist predators–consumed distinct prey; and the prey diversity was strongly correlated with the gut bacterial profile. Such host dietary specialization and spatial variation in bacterial communities suggested passive rather than selective underlying processes. Indeed, the abundance and distribution of 72% of bacterial taxa were consistent with neutral community assembly; and fluorescent in situ hybridization revealed that bacteria only rarely colonized the gut lining. Our results contradict the expectation that host-imposed selection shapes the gut microbiota of most insects, and highlight the importance of joint analyses of diet and gut microbiota of natural host populations.

Introduction

Host-associated gut microbial communities can have large impacts on host evolution. In turn, the gut microbiome is affected by many factors including host genotype, environmental variation, and host diet (Broderick & Lemaitre, 2012; Colman, Toolson & Takacs-Vesbach, 2012; McFall-Ngai et al., 2013; Kostic, Howitt & Garrett, 2013; Engel & Moran, 2013). However, it is not always clear whether these effects of host genotype, diet and environment reflect variation in the acquisition or the establishment step of microbial community assembly. Gut microbes are typically acquired from the mother, through social contact with conspecifics, or the diet; and they may either colonize the gut or fail to establish. At each step, various stochastic vs. deterministic, and neutral vs. selective processes determine community composition. For instance, a host may consistently acquire a specific set of microbes if they are maternally transmitted, or if the host is a dietary specialist. Within the host gut, microbial survival and growth dynamics may be primarily determined by stochastic neutral processes (e.g., based on initial abundance); or by deterministic and selective processes such as interactions with the host or with other microbes.

Figure 1 (A) Map of India showing dragonfly sampling locations (Map of India: ©Anuradha Joglekar and Krushnamegh Kunte, NCBS). Sampling details are given in Table S1. (B) Orthetrum pruinosum. (C) Othetrum sabina, (D) Pantala flavescens, (E) Trithemis aurora, (F) Urothemis signata, (G) Zyonyx iris. (Photo credit: B–F: ©Dattaprasad Sawant, G: ©Krushnamegh Kunte, NCBS. (H) Major bacterial phyla and classes in the dominant gut bacterial communities of sampled dragonflies. Numbers in parentheses indicate the number of OTUs (dominant community) in each taxonomic group. (I) Heat map showing dominant bacterial OTUs across all dragonflies. Each column indicates a host individual (sorted by species), and rows indicate dominant bacterial OTUs clustered based on their abundance across hosts.

Gut-associated microbiomes can thus be considered as meta-communities (Sloan et al., 2007; Woodcock et al., 2007), where hosts represent local habitats that are colonized by microbes. Gut microbial community composition should thus depend on dispersal, subsequent filtering by the local environment (host), and successful establishment. Prior studies show that host diet can act as a major source of microbial colonizers, and thus can alter microbial dispersion into the host gut (Engel & Moran, 2013), impact nutrient availability (for the microbes) in the gut (Laparra & Sanz, 2010), and introduce new parasites thus inducing immune responses (Walk et al., 2010). In general, it is expected that generalist hosts are more likely to stochastically sample a wider range of environmental microbes associated with a variable diet (in comparison with dietary specialists), as observed for scavengers and omnivores (Yun et al., 2014; Yadav et al., 2015; Shukla et al., 2016). This disruptive effect of dietary variation can be opposed by the strong host-imposed selection, stabilizing gut bacterial community composition. Many studies have implicated such host-imposed selection as a driver of gut bacterial community composition (Spor, Koren & Ley, 2011; Engel & Moran, 2013; Antwis et al., 2017). For instance, host immune responses (Ley, Peterson & Gordon, 2006; Charroux & Royet, 2012; Broderick & Lemaitre, 2012; Quigley, 2013) or a host-derived protected niche inside gut crypts (Dillon & Dillon, 2004; Kikuchi, Hosokawa & Fukatsu, 2007; Engel & Moran, 2013) may allow only specific microbes to colonize the gut. In contrast, under weak host selection, neutral processes such as ecological drift and microbial dispersal may strongly drive community assembly (Hubbell, 2001; Rosindell, Hubbell & Etienne, 2011), with each host’s microbiota functioning as a local community interacting with the larger meta-community outside the host (Costello et al., 2009; Costello et al., 2012).

Although several studies have found substantial individual and population level variation in the microbiomes of wild-collected insects (Osei-Poku et al., 2012; Martinson, Douglas & Jaenike, 2017; Sanders et al., 2017; Adair et al., 2018), the relative importance of various environmental and host-specific factors in determining the composition and stability of gut bacterial communities of natural animal populations remains unclear. We analyzed the gut bacterial and prey community composition in natural populations of six dragonfly species (adults), sampled from five locations in India (Fig. 1A, and Table S1) without significant temporal variation (each location sampled within 3–4 days). Dragonflies are generalist predators (Corbet, 2004) (i.e., predating on a diversity of prey items across its distribution); thus, we expected that they would consume diverse insect prey across locations and host species. In turn, this dietary diversity should be associated with diverse gut microbial communities. Previously, we found that the culturable fraction of gut bacterial communities of adult dragonflies varied significantly as a function of host species, location, and sampling time (Nair & Agashe, 2016). Here, we built upon this work by sampling more dragonflies, identifying most gut-associated bacteria using 16S rRNA amplicon sequencing, and analyzing host diet by sequencing the cytochrome c oxidase 1 gene (COX1) from gut contents. We then quantified the spatial stability of host-associated gut microbiomes; tested whether bacterial diversity correlates with host diet diversity; and quantified the relative importance of neutral processes driving bacterial community assembly.

Materials and Methods

Sample collection and storage

We collected adult dragonflies from five different sampling sites across India (Fig. 1A and Table S1A), focusing on the species Orthetrum pruinosum, Orthetrum sabina, Pantala flavescens, all from the family Libellulidae, order Odonata (http://indiabiodiversity.org/ for species identification). We also sampled three other dragonfly species Trithemis aurora, Urothemis signata, and Zygonyx iris (family: Libellulidae), from one geographic location—Shendurney (Fig. 1A, Table S1B) (for shendurney samples only permit obtained from Kerala Forest Department, permit no. WL 10-3781/2012 dated 18/12/2012 , and GO (RT) No. 376/2012/F and WLD dated 26/07/2012). We mainly sampled during the monsoon and post-monsoon season (October— early January) when dragonfly populations are at their peak (except for three individuals of species O. sabina that were collected during summer from Agumbe region). We sampled each location once within a span of 3–4 days during the monsoon and post monsoon (Table S1). We caught individuals using butterfly nets in open grounds, near natural water bodies, or waterlogged paddy fields. We collected three sets of samples to determine (a) the composition of gut bacterial communities (using amplicon sequencing of 16S rRNA), (b) the absolute abundance of focal bacteria (using qPCR) and to localise them in the gut (using fluorescent in situ hybridization, FISH), and (c) dragonfly diet (using amplicon sequencing of cytochrome c oxidase 1 gene (COX1)).

(a) To determine the composition of gut bacterial communities, we surface sterilized each dragonfly (see Table S1 for sampling details) using 70% ethanol and stored it in a 1 × 1 ft. mesh cage. Within 4–6 h of collection, we paralyzed dragonflies using a 4 °C cold shock and dissected them in phosphate-buffered saline (PBS) using sterilized dissection tools. We stored dissected guts in 1.5 ml centrifuge tubes containing 100% molecular grade ethanol. We stored the remaining dragonfly bodies separately in 100% ethanol for subsequent identification based on a morphological key using an online resource (http://indiabiodiversity.org/). After bringing samples to the laboratory, we stored samples at −20 °C until further processing. For collections in Bordubi and Nagpur, we could not dissect dragonflies in the field, so we stored them in 100% ethanol immediately after capture. We dissected these samples also within 4–6 h of capture.

(b) To estimate absolute gut bacterial abundance (using qPCR) and to localize bacteria in dragonfly guts (using FISH), we collected and isolated dragonflies in 50 ml Falcon tubes for 4–6 h (see Table S1 for sampling details). Such isolation allowed the dragonflies to defecate and remove large and probably indigestible food particles from the gut, which can cause difficulties during microtome sectioning of gut tissues. For dragonflies collected for qPCR, we dissected the entire gut in PBS and stored the guts in 100% ethanol. For FISH, we dissected the gut in PBS, divided each gut into three sections (foregut, midgut, and hindgut), and stored each section separately in 100% ethanol. We brought the dragonfly bodies and the guts back to the laboratory and stored them at −20 °C.

(c) To analyze the dragonfly diet, we again collected individuals of three of the well-sampled dragonfly species used for gut bacterial community analysis (O. pruinosum, O. sabina, and P. flavescens) (Fig. 1A, see Table S1 for sampling details). We dissected out the entire guts in PBS. We made an incision in the gut wall and centrifuged the gut for 30 s at 1000 rpm to separate gut contents from the gut tissue. We only stored the gut contents in Eppendorf tubes in 100% ethanol. We brought the dragonfly bodies and the guts back to the laboratory and stored them at −20 °C until further processing.

Amplicon sequencing to determine gut bacterial and diet composition

We determined the gut bacterial community for a total of 47 dragonflies from different species and geographical locations (Fig. 1A, Table S1). We washed each stored gut sample (intact gut along with its contents) thrice in fresh 100% molecular grade ethanol followed by three washes in PBS. We homogenized the tissue in liquid nitrogen using single-use sterile pestles and extracted DNA using the Wizard® Genomic DNA Purification Kit (Promega Corporations, Madison, WI, USA). We modified the manufacturer’s protocol as follows: we added 600 µl of nuclei lysis solution (10 mM EDTA) per 100mg tissue and incubated first at 80 °C for 20 min, and then at 65 °C for 30 min. We cooled the samples to 55 °C, added 20 mg/ml proteinase, and again incubated at 55 °C for 3 h. To precipitate degraded protein, we added protein precipitation solution and left the sample on ice for 30 min. We centrifuged the lysate at 14,000 g for 10 min and precipitated the supernatant with isopropanol. We washed the resulting pellet with 80% ethanol twice, then dried and suspended it in 40 µl ultrapure nuclease-free water. We quantified DNA in a Nano-Drop (Nano-drop 2000, Thermo Fisher Scientific Inc., Wilmington, USA). We also generated negative controls, where we prepared 16S rRNA libraries without adding sample DNA, and quantified DNA after each PCR step using a highly sensitive Qubit 3 fluorometer (Invitrogen, ThermoFisher Scientific Inc.) (as described in Phalnikar, Kunte & Agashe, 2018). However, we did not detect any DNA in any of the blank samples; in contrast, we detected high concentrations of DNA (85–300 ng/µl) for all gut microbe samples. We also checked the integrity of the DNA (for gut samples) by running 1 µg on a 0.8% agarose gel. For each sample, we used 50 ng DNA to PCR-amplify the V3-V4 hypervariable region of the bacterial 16S rRNA gene, using ExTaq (TaKaRa). The PCR primers contained tag sequences complementary to the Illumina sequencing adapter and index primers from the Nextera XT Index kit V2. We tested amplicons for quality and sequenced them (250 bp paired-end) on the Illumina MiSeq platform (Illumina, San Diego, CA, USA) using standard Illumina forward and reverse primers. Sequencing was performed by Genotypic Technology Pvt. Ltd., Bangalore, India.

For host diet analysis, we implemented (with few modifications) a previously described method that was used to estimate diet diversity in insectivorous bats (Zeale et al., 2011), and recently in odonates (Kaunisto et al., 2017). A recent study by Kamenova and colleagues (2017) showed that prey DNA remains relatively intact inside the gut of a predatory carabid beetle (Pterostichus melanarius) for at least 3–5 days. Assuming a similar prey retention time in dragonfly guts, we thus expected that our analysis would reflect a 3–5 day snapshot of dietary diversity in each dragonfly. In brief, we targeted the variable region of the COX1 gene—found in all insects— to estimate insect prey diversity from the gut contents of captured dragonflies. Prior studies suggest that COX1 primer sites (as used in Zeale et al. (2011)) are often less conserved, leading to unreliable amplification, especially in animal groups such as nematodes (Deagle et al., 2014). However, currently COX1 has the best database of barcode sequences with taxonomically verified organisms, which is critical for diet studies. We sampled a total of 45 dragonflies representing three species; as well as a phytophagous butterfly larva (Hasora sp.) as a positive control. The phytophagous butterfly provided an ideal control, since we expected amplification of only the host DNA in this sample. Any non-host reads would indicate non-specific amplification of contaminants during sequencing. We expected that if our classification were correct, it would classify all the sequences coming from this sample as Hasora sp. We extracted dragonfly gut contents after removing the host tissue (as described earlier) and then extracted DNA from the gut contents. For our control sample, we homogenized the tissue in liquid nitrogen using single-use sterile pestles and used then further processed it for DNA extraction. We extracted DNA using the Wizard® Genomic DNA Purification Kit (Promega Corporations, Madison, WI, USA) with the following modifications. We lysed cells at 65 °C in nuclei lysis solution with 10 mM EDTA, followed by an overnight proteinase K treatment. We precipitated DNA overnight at −20 °C, suspended the final pellet in 20 µl nuclease-free water, and checked the concentration and integrity of the DNA. For further analysis, we chose samples showing intact bands on an agarose gel (n = 28 dragonflies, and 1 butterfly larva; Table S1). We designed custom primers—containing Illumina ITS barcodes for multiplexing—to target the COX1 variable region (using references from Zeale et al., 2011). The forward primer sequence was 5′-TCGTCGGCAGCGTCAGATGTGTATAAGAGACAG—AGATATTGGAACWTTATATTT TATTTTTGG-3′, and the reverse primer was 5′-GTCTCGTGGGCTCGGAGATGTGTA TAAGAGACAG- WACTAATCAATTWCCAAATCCTCC-3′. We used 200 ng DNA from each sample to amplify the target COX1 region with High Fidelity Phusion polymerase (Thermo scientific). We purified the samples using the Qiagen PCR purification Kit (Qiagen) and checked the product for amplicon size and concentration (as mentioned above). We did not use blocking primers (Vestheim & Jarman, 2008; De Barba et al., 2014) or PNA (thermally more stable) (Chow et al., 2011; Terahara et al., 2011) to amplify prey DNA because we did not know the prey diversity in dragonflies, and some dragonflies are known predators of other (closely related) dragonflies. These two factors made it difficult to design blocking primers or PNA that would block only host DNA (Piñol et al., 2014). We also prepared blank samples (as negative controls) and tested for DNA concentration in these blank samples to check for possible contamination (as described above). Concordant to our previous extraction, we did not find any detectable DNA concentration in our blank samples. Finally, for the dragonfly gut content and the positive control (Hasora sp. larva) sample, we prepared sequencing libraries using the Nextera XT v2 Index Kit (Illumina, U.S.A.) and sequenced them on the MiSeq platform (250 bp paired-end). Sequencing was performed by Genotypic Technology Pvt. Ltd., Bangalore, India.

We processed amplicon sequencing data using QIIME (Caporaso et al., 2010) using conservative protocols to reduce errors. After demultiplexing and removing barcodes and primer sequences, we filtered and trimmed reads for sequence length and quality score (q>20) using default QIIME parameters. We used Fast-QC (Babraham Bioinformatics, 0000) to check read quality and presence of barcodes or primers in the processed data (see Fig. S1 for workflow). Finally, we paired the forward and reverse reads to generate a total of 30 million high-quality paired-end reads for the 16S rRNA gene, with an average of 169,000 reads per sample (range: 8,000–900,000, post quality filtering 62,000–599,000, Fig. S2). We classified reads into Operational Taxonomic Units (OTUs) at the 97% similarity level using UCLUST (Edgar, 2010), using both open and closed reference OTU picking. For OTU picking, we set both “maxaccepts” and “maxrejects” values to 0, to ensure an exhaustive search. We used the GreenGenes 16S rRNA ribosomal gene database version gg_13_8 (DeSantis et al., 2006) to assign taxonomy to each representative OTU. We removed chimeric sequences using Chimeraslayer (Haas et al., 2011) and also removed unassigned, chloroplast, and mitochondrial sequences to generate the final “.biom” files for OTU picking. We normalized closed referenced OTUs by bacterial 16S rRNA copy number using the software PICRUSt (Phylogenetic Investigation of Communities by Reconstruction of Unobserved State, version 1.0.0) (Langille et al., 2013). For insect COX1 amplicons, we obtained a total of 2.1 million reads (average 70,000 and range 29,000–115,000 reads per sample, Fig. S2). We picked OTUs using similarity cut off (97%) described in Hebert, Ratnasingham & De Waard (2003) for COX-1 region and used the Barcode of Life Database v4 (Ratnasingham & Hebert, 2007) (Chamberlain, 2019) to assign taxonomy to each OTU. We removed chimeric sequences and checked the precision of our sequencing and OTU assignment using our control sample (Hasora sp. butterfly larva), where 97.4% of the reads were correctly classified to a single OTU assigned to Hasora sp. We removed the most abundant Odonate OTU from each sample since it likely represented host amplicons, although this could also remove potential cases of conspecific predation (Corbet, 2004) (Fig. S3). However, this step was essential because several studies examining host diet using host gut or in faecal samples have reported high proportion of host DNA in their sequencing output (Piñol et al., 2014; De Barba et al., 2014; Kaunisto et al., 2017).

We removed potentially erroneous OTUs (as described in Huse et al., 2010) from both amplicon datasets by implementing three OTU filters. (1) Pruned community: retaining all OTUs with at least 0.005% relative abundance across the entire dataset, to minimize impacts of sequencing errors (Bokulich et al., 2013); (2) Dominant community: retaining all OTUs with at least 5% relative abundance in at least one sample; (3) Minimally pruned community: retaining all OTUs with at least 20 reads per OTU per sample, to obtain a conservative estimate of OTUs with sufficient read support. We separately applied each filter to the full dataset and then recalculated the relative abundance of OTUs for subsequent analysis.

Statistical analysis

We performed all statistical analysis in the R statistical software version 3.3.4 (R Core Team, 2013), considering each OTU as the basic unit of comparison regardless of taxonomic placement. To estimate the sampling depth at which the community richness saturated, we performed a rarefaction analysis with the Pruned Community in QIIME (Caporaso et al., 2010). We subsampled reads to simulate varying sampling depth (100–2,500 reads per sample) and calculated Faith’s phylogenetic diversity (Faith’s PD) at each depth. We then estimated the sampling depth at which PD saturated, as an indicator of sufficient sampling. We also rarefied the community to its lowest sampling depth and subsampled the community 100 times. We calculated the mean rarefied community and its standard deviation, and also used these data for analysis. However, in the results section, unless specifically mentioned all analysed communities are unrarefied communities.

We analyzed community structure (relative abundance of OTUs) across samples using Ward’s hierarchical agglomerative clustering (Murtagh & Legendre, 2014). We tested the impact of host species and geographical location on the gut bacterial community composition using permutational ANOVA (PERMANOVA, in the R package vegan (Dixon, 2003; Oksanen et al., 2017) using function adonis) with 10,000 permutations. We also performed PERMANOVA using library size as an additional factor, to understand whether library size influenced the outcome (Weiss et al., 2017). We used the R package “Caret” (Kuhn, 2008) to remove near-zero variance in the data (i.e., OTU vectors with little or no variance). We performed this step because these vectors provide no information, but did not allow our downstream analysis (CAPdiscrim function) to proceed. To visualize sample clustering based on bacterial composition across treatments, we calculated Bray-Curtis distances between samples and performed Canonical Analysis of Principal Coordinates based on Discriminant Analysis (CAPdiscrim Anderson & Willis, 2003) using the R package “Biodiversity R” (Kindt, 2019). We tested the significance of clustering and estimated classification success by permuting the distance matrix 1000 times and calculated P value. We plotted the two dominant linear discriminants (LD) to visualize data classification. For each cluster, we drew ellipses reflecting 95% confidence intervals using the function “Ordiellipse” in the R package “Vegan” (Dixon, 2003; Oksanen et al., 2017). This analysis was also perform on the mean rarefied community for comparison.

To estimate bacterial or prey OTU richness for each dragonfly sample, we generated a presence-absence matrix from the relative abundance of each OTU. We used the final “.biom” table to identify shared OTUs across samples, and to calculate OTU richness per sample, and α diversity (Shannon’s diversity index, a measure of OTU richness and evenness per sample), βw (a comprehensive measure of the number of unique OTUs per sample Koleff, Gaston & Lennon, 2003) using the R package Vegan (function “diversity” for Shannon’s index, and “betadiver” for βw diversity) (Dixon, 2003; Oksanen et al., 2017) and βAitchison diversity (measure of β-diversity for compositional data using R-package robCompositions (Martino et al., 2019)). We also measured the α diversity and βw for mean, +1SD and −1SD rarefied communities for comparison. We tested the effect of host species identity, and sampling location on bacterial OTU richness using a generalized linear model (GLM) with Poisson errors (using the R package Stats R Core Team, 2013). We also performed Kruskal Wallis’ test followed by post-hoc Conover test (using the R package PMCMR (Pohlert, 2018) to compare between bacterial richness and diversity with the prey richness and diversity of the three main dragonflies.

Testing models of bacterial community assembly in dragonfly guts

If the gut community of a host is under weak selection, we expected it to be predominantly neutrally assembled. To test whether a neutral model of community assembly could explain the observed distribution of bacterial communities across hosts, we fitted a neutral distribution model (Sloan et al., 2006; Woodcock et al., 2007) to the bacterial communities observed in dragonfly hosts from a specific location. The model is based on Hubbell’s model of the neutral theory of biodiversity (Hubbell, 2001), but applies to large communities, such as a complex microbial community. For model fitting (using the open reference pruned community), we followed the approach used by Burns et al. (2016). We assumed that each individual dragonfly gut houses a local community with numerous bacterial species (OTUs) whose members are drawn from a larger metacommunity, comprised of bacteria present across all dragonfly individuals collected from a specific geographic location. The model uses the following parameters: (a) population size of each OTU in the local and metacommunity (estimated using the number of reads) (b) the relative abundance of each OTU. Using these, the model estimates the migration rate or dispersion probability (m) for each OTU. In the event of an individual bacterium’s death, m is the probability that it will be replaced via dispersal from the metacommunity, rather than reproduction within the local community. The relationship between the abundance of each OTU in the metacommunity and its occurrence across local communities is informative for understanding the processes driving community assembly (Sloan et al., 2006). For each metacommunity, we fit a β-distribution to the relationship between OTU occurrence and abundance (using the script published by Burns et al. (2016); please see Supplementary Code 1 published in Burns et al., 2016 (https://media.nature.com/original/nature-assets/ismej/journal/v10/n3/extref/ismej2015142x4.txt); R-packages used: vegan (Dixon, 2003; Oksanen et al., 2017), minpack.lm (Elzhov et al., 2010), Hmisc (Harrell Jr & Dupont, 2008). We checked the fit using non-linear least squares in R and estimated 99% (and 95%) confidence intervals (CI) around the fit using binomial proportions. We then compared the proportion of OTUs that were neutrally distributed across sites and hosts. Under neutral assembly, a highly abundant OTU should occur in many hosts. If an OTU occurs at a higher frequency in a host than expected from its abundance in the metacommunity (comprised of OTUs from all hosts), this indicates positive selection for those bacteria (presumably by the host). Similarly, if an OTU is very abundant in the metacommunity but occurs in only a few host individuals, this indicates negative selection against the OTU. Finally, we compared taxonomic diversity (Clarke & Warwick, 1998; Fierer, Bradford & Jackson, 2007; Morrow et al., 2015) between the groups of bacteria that were inferred to be neutrally distributed or positively or negatively selected by hosts. If hosts selected for a specific functional association (and this functionality is phylogenetically conserved in bacteria), we expected that bacterial OTUs experiencing positive host selection should have lower taxonomic diversity compared to neutrally assembled bacteria. To estimate the fit of each model, we calculated generalized R square (R2) (as suggested by Burns et al. (2016)). We also fitted a binomial distribution model that assumed that the local communities were only a subset of the meta-communities without any dispersion and drift (as suggested by Burns et al. (2016) and Sloan et al. (2007)). We compared the fit of both binomial as well as Sloan’s neutral models using R2 and Akaike information criterion (AIC) (as suggested by Burns et al. (2016)).

We generated two sets of models; (1) for each dragonfly species sampled at a specific location, and (2) pooling all dragonfly species sampled in each location. The first set allowed us to infer patterns of gut bacterial community assembly for each dragonfly species, but with low sample sizes (Table S1). The second set allowed us to infer general patterns of gut bacterial assembly across dragonflies, with a larger sample size.

Localizing bacteria in dragonfly guts

We used fluorescent in-situ hybridization (FISH) to determine the location of bacteria inside dragonfly guts. We hypothesized that if there is a functional association between host and bacteria, bacterial cells should be housed in specific crypts or inside columnar cellular folds in the host gut (as reported in previous studies by Barrow et al., 1980; Fuller & Turvey, 1971). We performed FISH for three dragonfly species (O. sabina, O. pruinosum, and P. flavescens; n = 5 individuals per species). We used a universal eubacterial probe ([Alexa-488] 5′-GCTGCCTCCCGTAGGAGT-3′(Da Silva et al., 2015)) and a Wolbachia-specific probe ([ALEXA-647] 5′-CTTCTGTGAGTACCGTCATTATC-3′  (Le Clec’h et al., 2013)) (Sigma-Aldrich-Merck, Missouri, USA) to stain the specific gut bacteria. We used DAPI (4′, 6-diamidino-2-phenylindole) staining to visualize host cell nuclei. We followed the COLOSS protocol (Engel et al., 2013) with a few modifications. Before the assay, we rehydrated guts and fixed them in Carnoy’s fixative for 96 h. To reduce autofluorescence, we used peroxide treatment for 72 h and replaced the water inside host tissues using absolute ethanol and xylene washes as per the COLOSS protocol. Finally, we embedded samples in liquid paraffin using plastic moulds to create paraffin blocks. We sliced the blocks into 10  µm transverse sections using a Leica manual microtome (Leica Microtome 2125 RTS, Wetzlar, Germany) using disposable blades (Low profile blade 819, Leica). For each species and each part of the gut (foregut, midgut, and hindgut), we obtained 5 sections per probe for each of 5 individuals. We mounted sections on Fisherbrand Superfrost Plus microscope slides (Thermo Fisher Scientific, Wilmington, USA), and heated off the paraffin in an oven at 65 °C. We washed with Xylene (three-minute wash, thrice), absolute ethanol (three-minute wash, thrice), and double-distilled water (once) before hybridization. We dissolved 0.5 µL of fluorescent probes in 500 µL hybridization buffer and stained gut tissue sections in a dark chamber for 8–10 h at room temperature with the respective bacteria-specific probe. We then stained sections with DAPI for 20 min to visualize host cell nuclei. We applied DABCO-glycerol (antifade agent), sealed the sections with coverslips, and stored them at 4 °C in the dark. For each species and each part of the gut (foregut, midgut, and hindgut), we obtained 5 sections per probe for each of 5 individuals. Though we lost many sections during the multiple washes, we retained at least 2 sections/dragonfly/species/probe for final analysis. We imaged sections using a Zeiss 510 Meta confocal microscope (Oberkochen, Germany) and analyzed images using Image-J software (Version 1.6.0-24, 64-bit version). As a positive control, we probed heat-fixed bacterial smears (5 slides of Bacillus thuringiensis from laboratory culture) with the eubacterial probe (following the above protocol). As a negative control, we probed dragonfly tissue sections (3 midgut tissue sections of each host species) using fluorophore-free eubacterial and Wolbachia probes, before hybridizing with the probes containing active fluorophores. We expected that fluorophore-free probes would get attached to the bacterial cells and prevent the fluorophore-attached probes from binding with bacterial DNA.

Quantitative PCR to validate the abundance of specific gut bacteria

To estimate the abundance of eubacteria and Wolbachia (a common insect-associated bacterial genus), we performed quantitative PCR (qPCR) on bacterial DNA extracted from 9 dragonfly guts (three individuals each of O. sabina, O. pruinosum and P. flavescens). We used previously reported primers (Heddi et al., 1999): universal Eubacterial primers, forward 5′-AGAGTTTGATCATGGCTCAG-3′ and reverse 5′-TACCTTGTTACGACTTCACC-3′; and Wolbachia specific primers, forward 5′-CGGGGGAAAAATTTATTGCT-3′, reverse 5′-AGCTGTAATACAGAAAGTAAA-3′. To normalize bacterial abundance to host tissue, we used previously described Odonate-specific primers for the 28S gene (forward: 5′-ACCATGAAAGGTGTTGGTTG-3′ and reverse: 5′-ATCTCCCTGCGAGAGGATTC-3′) (Dijkstra et al., 2014). All primer pairs had amplification efficiencies greater than 90%. We ran three sets of PCR for each sample (total 10 µL reaction volume), using 10 ng of host gut DNA, 8 µL SYBR green PCR master mix (Thermo Fisher Scientific, Wilmington USA), and the appropriate primers (200 nM each). We added reaction mixes in 384 well microplates (Corning, New York, USA) and monitored amplification in a ViiA™ 7 Real-Time PCR System (Thermo Fisher Scientific, Wilmington USA). We used the following cycle conditions: 95 °C for 30 s, 40 cycles of 95 °C for 60 s, 56 °C for 60 s, 72 °C for 60 s, and extension at 72 °C for 5 min. We calculated threshold cycle values (CT) for each sample. We used the CT value of each host-specific gene to estimate the ΔCT values. Finally, we plotted these values for all the three host dragonflies for comparison.

Results

The gut bacterial community of dragonflies

Initial rarefaction analysis (collector’s curve) revealed that our sampling depth was sufficient to determine the bacterial community composition in all but one sample (one Orthetrum sabina sample), which we excluded from further analysis (Fig. S4). Our PERMANOVA analysis revealed that library size did not influence the gut bacterial community composition (Tables S2A and S2B). As all samples showed sufficient sampling depth, we used the original dataset (non-rarefied) for our entire analysis. Moreover, McMurdie & Holmes (2014) have shown that rarefied datasets can lead to an erroneous measurement of species abundance, especially if a species is differentially abundant across sample communities. However, in relevant sections, we have also highlighted the results of our analysis of the mean rarefied (mean of 100 iterations) community for comparison. We separately analyzed a total of six sets of non-rarefied bacterial communities across all dragonfly species (6) and locations (5), generated using either closed or open-reference OTU picking and implementing three OTU filtering thresholds: (1) pruned community (576 OTUs open reference), (2) dominant community (59 OTUs open reference), and 3) minimally pruned community (2599 OTUs open reference). All sets showed comparable results, but here we focus on the pruned and dominant open referenced sets unless mentioned otherwise. Corresponding results for other sets as well as rarefied communities are given in the Supplementary Material.

We found an average of 188 OTUs per sample, most belonging to Proteobacteria (88%), Firmicutes (9.8%), Actinobacteria (1.8%) and Bacteroidetes (0.4%) (Figs. 1B & 1C). At the family level, Rickettsiaceae—comprising of three Wolbachia OTUs—were most abundant, although this high abundance was limited to dragonflies from the genus Orthetrum (Fig. 1C). In other host genera, especially P. flavescens, OTUs from the family Enterobacteriaceae were more abundant. Overall, we observed substantial variation in the relative abundance of OTUs across host species, as well as across host individuals (Fig. 1C, Fig. S5: all species, open referenced, dominant community).

Host species and sampling location shape gut bacterial community composition

We found that both host species and sampling location significantly affected the composition of the dominant gut bacterial community (PERMANOVA; Table 1; see Table S3 for other community sets) across the three best-sampled dragonflies (Table S1A). Linear discriminant analysis (constrained by the group, CAPdiscrim) to visualize clustering supported these results, showing strong separation in bacterial communities across host species and sampling location (Figs. 2A–2B, S5, and S6; see Figs. S7A–S7B for mean rarefied community). Interestingly, location explained a considerably larger proportion of variation in gut bacterial communities (26%; Table 1, Table S3) compared to host species alone (9%), suggesting that environmental factors (such as local microbial community and prey community, soil pH, or rainfall) have a stronger impact on community composition (Figs. 2A–2B, S6A–S6B, S7A–S7B). This pattern was consistent irrespective of the OTU filtering or referencing methodology used (open or closed; Table S3). When we restricted our analysis to the geographic location Shendurney, where we had sampled a total of 5 different dragonfly host species (Table S1B), we found a similarly weak yet significant impact of host species on dragonfly gut bacterial composition (Table 1B and Fig. 2C; Fig. S7C for rarefied mean community).

Table 1 (A) Results of a permutational analysis of variation (PERMANOVA) showing the effect of host species and location on gut bacterial composition (open referenced dominant bacterial community) across the three best-sampled dragonfly hosts (Orthetrum pruinosum, Orthetrum sabina, and Pantala flavescens). (B) Results of PERMANOVA showing the individual comparisons of 5 host species (Orthetrum pruinosum, Pantala flavescens, Trithemis aurora, Urothemis signata, and Zygonyx iris) based on their gut bacterial composition at a single geographic location—Shendurney.

(A)	Df	SSq.	Mean SSq.	F stat	R2	P	
Species	2	1.31	0.66	2.36	0.09	0.005	
Location	4	3.77	0.94	3.40	0.26	9.9 × 10−5	
Interaction (Species, Location)	3	1.41	0.47	1.69	0.10	0.032	
Residuals	29	8.03	0.28		0.55		
Total	38	14.52			1		
(B)	Estimate	Std. error	t value	P	
Intercept	0.11	0.09	1.15	0.28	
Species (P. flavescens)	−0.26	0.12	−2.09	0.07	
Species (T. aurora)	−0.47	0.12	−3.84	0.005	
Species (U. signata)	0.33	0.13	2.47	0.04	
Species (Z. iris)	0.03	0.12	0.25	0.81	
Notes.

Residual standard error: 0.13, df = 8, Adj. R2 = 0.80, F4,8 = 12.9, P = 0.001.

These patterns were also mirrored in the number of shared bacterial taxa across dragonflies. Of the 576 OTUs detected in total across all dragonflies, 206 OTUs (∼36%) were found in all host species (Fig. S8E). The percentage of shared OTUs increased to 64% (366 OTUs out of 571 OTUs) when we considered only the three well-sampled dragonflies (Table S1A). The congeneric dragonflies O. pruinosum and O. sabina, which harbored similar bacterial communities (Fig. 2A), also shared the maximum number of bacterial OTUs (407 shared OTUs, ∼71%, of which 34 OTUs were unique to the genus Orthetrum; Fig. S8E). Only 25% of the OTUs (145 out of 576 OTUs) were shared across locations (Fig. S8F) when we considered all dragonfly samples. This proportion increased marginally (30%, 175 out of 575 OTUs) for the subset of well-sampled dragonflies (Table S1A). Finally, for each explanatory factor, classification analysis based on gut bacterial composition categorised significant proportions of samples correctly into the respective groups (for well-sampled dragonflies: Figs. 2A–2B, Tables S4A–S4B (open reference dominant community) and S5A–S5B (closed reference dominant community); for 5 dragonfly species at Shendurney: Fig. 2C, Tables S4C (open reference dominant community) and S5C (closed reference dominant community)).

Figure 2 Linear discriminant (LD) plots showing two dominant linear discriminants (LD) that group dragonfly samples based on their gut bacterial community composition (based on Bray-Curtis distance and open reference OTU picking).

Percentage of variance explained by each LD is indicated in parentheses. Each point represents a host individual. Ellipsoids represent 95% confidence intervals around each group mean, calculated from LD values. Clustering of dragonfly samples based on (A) host species identity (3 well-sampled species) (Table S1A), (B) sampling location (for 3 well-sampled species) (Table S1A). (C) Clustering of 5 dragonfly species from Shendurney (Table S1), based on their gut bacterial composition.

Our amplicon sequencing data had revealed that families Rickettsiaceae and Enterobacteriaceae were most abundant in the gut of dragonflies (Fig. 1C). We tested whether removing these two dominant families altered the impact of host species and sampling location on the observed community composition. We found that the results were robust to their removal (Fig. S12). Conversely, the abundance of OTUs from these families was influenced both by host species and sampling location (Fig. 1C and Fig. S13; Tables S6 and S7), mirroring our results for the full dataset. Interestingly, we found an extremely high abundance of bacteria from the family Rickettsiaceae in the gut tissues of samples from the genus Orthetrum (Fig. 1C, Fig. S13). We validated this observation using qPCR, which revealed that Wolbachia (which belongs to this family) was much more abundant in the genus Orthetrum compared to P. flavescens (Fig. S14).

Despite the significant effects of host species and location on community composition (Fig. 2, S6), these factors had relatively weak and variable impacts on bacterial community richness (Figs. S15A–S15C; Table S8A). The α-diversity of communities (considering both OTU richness and evenness) varied only across host species (Figs. S8A–S8B; see Fig. S9 for rarefied open reference communities); Table S8B and S9); whereas both factors significantly affected the β diversity (Figs. S8C–S8D; see Figs. S10 and S11 for rarefied open reference communities; Table S8C), indicating significant community turnover across species and sampling location. Interestingly, β diversity was higher in dragonflies collected from sites in Southern India (Agumbe, Bangalore, and Shendurney) compared to North Indian locations (Bordubi and Nagpur) (Figs. S8D, S10B). These results show that host-specific and environmental factors together govern bacterial community composition and turnover, with the latter having larger impacts.

Dragonflies show host-specific dietary specialization

To determine whether host-specific bacterial communities reflect host-specific diets, we next tested for dietary specialization across the three best-sampled dragonfly species, O. pruinosum, O. sabina, and P. flavescens collected from Agumbe region. We obtained 12.4% prey reads, out of a total of 2.1 million reads (Fig. S3). Our PERMANOVA analysis revealed that library size did not influence the prey composition in dragonfly guts (Tables S2C and S2D).

The pruned prey communities of both Orthetrum species had significantly higher richness compared to P. flavescens (P < 0.01, Kruskal Wallis’ Chi-square = 16.91, df = 2, post-hoc Conover test: OP vs. PF: P < 0.01, OS vs. PF: P < 0.01, OP vs OS: P = 0.04), as well as greater diversity (P = 0.003, Kruskal Wallis’ Chi-squared: 11.41, df = 2, post-hoc Conover test: OP vs. PF: P = 0.04, OS vs. PF: P = 0.0008, OP vs OS: P = 0.03) (Figs. 3A–3B; see Figs. S16A–S16B for rarefied mean community). These patterns mirrored the bacterial communities associated with these hosts (Figs. 3A–3B, Figs. S16A–S16B): Orthetrum had higher bacterial diversity (Kruskal Wallis’ Chi-squared: 7.39, df = 2, P = 0.02, post-hoc Conover test: OP vs. PF: P = 0.03, OS vs. PF: P = 0.04, OP vs. OS: P = 0.69) (Fig. 3A, Fig. S16A), though not significantly higher richness (Kruskal Wallis’ Chi-squared: 1.99, df = 2, P = 0.3) (Fig. 3B and S16B, though the trend was similar). The correlated differences in prey and gut bacterial communities suggest that they may be causally linked.

Figure 3 Boxplots show (A) Shannon diversity and (B) OTU richness of bacterial and prey communities of three dragonfly species. Asterisks indicate significant differences in richness (Kruskal Wallis test). (C) Clustering of dragonfly samples based on dietary composition using LD analysis, as described in Fig. 2.

Notably, we observed striking differences between the diets of the three dragonfly species (Figs. 3A–3B and Fig. S17; 2A–2B). Orthetrum pruinosum and O. sabina had similar diets composed predominantly of Dipterans (83% and 68% respectively), whereas P. flavescens consumed more Odonates (88% of prey OTUs, potentially indicating conspecific predation) (Fig. S17). This was also evident in our PERMANOVA results, which revealed that host species was an important factor driving prey composition (Table 2A). Finally, our classification analysis (CAPdiscrim) based on prey composition also categorized significant proportions (75%) of samples correctly into the respective groups (Table 2B), highlighting host specific dietary patterns in dragonflies. Since all the dragonflies collected for dietary analysis were collected from Agumbe region, we examined the gut bacterial composition of dragonflies (of the three well-sampled species) collected specifically from this area. Concordant to our previous analysis, we found a strong correspondence between the effect of host identity on dietary as well as the bacterial composition of dragonflies (Fig. 3C and Fig. S16C vs. Fig. S18; Table 2 vs. Table S10). Similarly, CAPdiscrim analysis also showed comparable classification success for samples based on both gut bacterial composition (81%) and prey composition (75%).

Table 2 (A) Results of a permutational analysis of variation (PERMANOVA) showing the effect of host species on the diet of Orthetrum pruinosum, Orthetrum sabina, and Pantala flavescens. (B) The output of an ordination model (CAPdiscrim) testing the impact of host species on dragonfly diet diversity.

 	Df	SSq.	Mean SSq.	F stat	R2	P	
(A)	
Species	2	3.24	1.62	8.12	0.39	9.9 × 10−05	
Residuals	25	4.99	0.20	 	0.61	 	
Total	27	8.23	 	 	1	 	
(B)	
Classification success	75%	P = 0.004					
Proportion of trace	LD1	LD2					
	0.94	0.057					
Manova	DF	Pillai approx	F	Numerator DF	Denominator DF	P	
Host Species	2	1.367	12.43	8	46	2.78 × 10−09	
Residuals	25						

Overall, these results suggest that (a) O. sabina and O. pruinosum primarily consume small dipterans and lepidopterans (Fig. S17); (b) but their target prey community still differed substantially (Fig. 3C); and (c) P. flavescens is a specialized predator that predominantly targets other Odonates (Fig. 3C and Fig. S17). Thus, a host-specific dietary pattern can potentially introduce distinct bacteria into host guts, directly determining the observed host-specific gut bacterial communities of dragonflies.

Dragonfly gut bacterial communities are predominantly neutrally assembled

Next, we specifically tested whether dragonfly gut bacterial communities are acquired passively through the diet, with relatively weak host-imposed filters. We estimated the fraction of the bacterial community whose occurrence and abundance across hosts was consistent with neutral vs. non-neutral assembly, analyzing communities from all samples collected from a given location (regardless of host species). We found that a large fraction of bacterial OTUs was predicted to be neutrally assembled (mean 72 ± 0.08% with 99% confidence intervals, Figs. S19A–S19E; mean 63 ± 0.08% with 95% CI; ), i.e., whose distribution across hosts matched expectations from a model simulating assembly via random OTU dispersal. We also compared the fit of the Sloan models with that of binomial models (as suggested by Burns et al., 2016) using both R2 and AIC estimation. In each case, the Sloan model explained more variation in the data than the Binomial model (Table S11). This result suggested that the local communities are not just a random subset of the meta-communities and dispersal plays an important role in structuring the local communities. Beta diversity (variation in composition across local communities) is therefore better explained when we account for dispersal between local communities (i.e., Sloan’s model). The proportion of neutrally distributed gut bacteria varied across locations (Fig. 4 and Fig. S19). Dragonflies collected from Bordubi had the highest proportion of neutrally assembled bacteria (83%), while those from Nagpur had the lowest proportion of neutrally assembled gut bacteria (63%; Fig. 4 and Fig. S19). As expected, we observed the opposite patterns for OTUs whose distribution was consistent with positive or negative selection (Fig. 4). We found similar results when we rarefied samples to the lowest read count per sample (e.g., 74% OTUs were predicted to be neutrally assembled, compared to 72% for the full dataset), suggesting that the model fitting was not sensitive to different read depth across samples. Finally, pooling all OTUs predicted to be under positive selection (across locations), we found that their taxonomic diversity was either higher than or comparable to OTUs that were neutrally distributed or under purifying/negative selection (Fig. S20). This result suggests that it is unlikely that dragonflies impose strong positive selection favoring a specific, shared set of functionally important bacteria. However, it is also possible that any functional traits selected by the host are not phylogenetically conserved (Louca et al., 2018), resulting in positive selection for unrelated bacteria. Currently, we cannot distinguish between these alternatives.

Figure 4 Barplots show the proportion of bacteria whose distribution is consistent with positive selection, neutral assembly, or negative/purifying selection, for dragonflies sampled from a given location.

Since we had relatively low sample sizes for each dragonfly species in a given location, we had restricted our analysis (above) to pooled samples— i.e., we included data from all dragonflies in a particular sampling location. However, we also attempted to investigate host species-specific patterns of gut bacterial community assembly (Fig. S21). We found that host species and sampling location both had a significant impact on the proportion of bacteria that are neutrally assembled (Table S12A and Fig. S21A; also see Table S12B and Fig. S21B for bacteria under positive selection). The dragonfly P. flavescens (which hunts other Odonates, which are also carnivorous and can also house non-specific neutrally assembled gut bacteria) had a higher proportion of neutrally assembled gut bacteria, in comparison to O. sabina and O. pruinosum (that predates on insect orders which can have host specific gut microbiota).

Bacterial cells rarely adhere inside dragonfly guts

Finally, to test whether bacterial cells adhere to dragonfly guts or are housed in specialized structures, we dissected the guts of three species (O. sabina, O. pruinosum, and P. flavescens) and probed for bacteria using FISH (Fig. 5). Our positive controls (bacterial smears on slides) showed a strong eubacterial signal, indicative of proper binding (Figs. S22A–S22B); while our negative control (see methods) did not show a signal. However, we did not find any eubacterial signal in the foregut (Figs. 5B–5D), indicating that bacteria were either absent or rare in this part of the gut. Since we did not find a signal with the general eubacterial probe, we did not test foregut sections with the Wolbachia-specific probe. In P. flavescens, only the eubacterial probe showed a positive signal inside columnar folds (5 of 5 tested individuals; 3 with small patches of bacteria) (Figs. 5E and 5H); whereas Wolbachia was absent (Figs. 5K and 5N), corroborating our amplicon sequencing results. The midgut and hindgut of both Orthetrum species were positive for eubacterial and Wolbachia-specific probes (Figs. 5F–5P; all 5 tested individuals of each species), although the signal was weak and localized to a small cluster of bacteria in the gaps between columnar cellular folds. Interesting exceptions were observed in two O. sabina individuals where Wolbachia appeared to be sequestered within a specific tissue structure (Fig. 5L); the functional significance of this pattern requires further work. Overall, the lack of a clear signal of bacterial gut colonization suggests at best a weak relationship with the host.

Figure 5 Examples of Fluorescent in situ hybridisation (FISH) images of dragonfly gut sections using bacteria-specific probes.

Host cell nuclei are stained purple with DAPI, eubacteria are green, and Wolbachia is pink. Arrows highlight bacteria in each section. (A) Representative brightfield image of P. flavescens midgut section showing columnar cellular folds covering the gut lumen, and food particles in the lumen. (B–D) Foregut sections of P. flavescens, O. sabina and O. pruinosum. Note the lack of eubacterial or Wolbachia signal. (E–G) Midgut and (H–J) hindgut sections of each species, stained with a eubacterial probe. Note the strong eubacterial signal near the columnar folds of P. flavescens. (K–M) Midgut and (N–P) hindgut sections of each species, stained with a Wolbachia- specific probe. Note the lack of signal in P. flavescens, a weak signal in O. pruinosum, and a large globular structure with Wolbachia in O. sabina.

Discussion

Host selection is generally considered to be a strong force shaping the gut bacterial communities of animals (Colman, Toolson & Takacs-Vesbach, 2012; Engel & Moran, 2013; Yun et al., 2014) and is expected to stabilize communities (Lozupone et al., 2012; Coyte, Schluter & Foster, 2015; Foster et al., 2017) in the face of environmental variation. Here, we tested this prediction by analyzing host associated gut bacteria across spatially separated populations of six dragonfly species. Our key results contrast multiple findings from prior work: (a) adult dragonfly bacterial communities are twice as rich and diverse as other carnivorous insects, including Odonates (Jones, Sanchez & Fierer, 2013; Yun et al., 2014); (b) geographic location explain more variation in bacterial community composition than host species identity; (c) adult dragonflies have specialized diets that reflect patterns of variation in gut bacterial communities, and (d) the adult gut community is predominantly neutrally assembled with regard to host species, showing little evidence of the strong host selection reported for many other insects (Engel & Moran, 2013; Yun et al., 2014). Thus, our work highlights the importance of analyzing gut microbial communities of natural host populations in the context of natural variation in geography and host taxonomy.

Our findings corroborate previous studies showing a significant geographical structure in the microbiomes of well-studied animal species such as humans, flies, and bees (Corby-Harris et al., 2007; Turnbaugh et al., 2009; Costello et al., 2009). Our prior analysis of culturable gut bacteria in dragonflies had also shown a weak yet significant effect of sampling location on the gut bacterial community (Nair & Agashe, 2016). In this study, the presence of a strong geographic structure (Fig. 2B) despite incorporating long-distance migrants like Pantala (Hobson et al., 2012) further highlights the importance of environmental factors shaping the gut bacterial community composition. However, the dragonfly bacterial community structure is not strongly associated with the distance between sites (Fig. 1A, Figs. S8, and S10). For instance, dragonflies of the same species collected from relatively close sites— Bangalore, Agumbe, and Shendurney— had distinct gut bacterial community composition, suggesting that a combination of multiple locally-acting factors may drive the composition of site-specific gut bacterial communities (Figs. 1A, 2B, S8, and S10). These factors may include specific environmental conditions (e.g., temperature, precipitation, and soil pH) that drive variation in environmental microbes; site-specific variation in host imposed selection acting on similar environmental microbes; variation in insect prey communities driving differential dispersal into host guts; or local host diet specialization (Dillon & Dillon, 2004; Osei-Poku et al., 2012; Engel & Moran, 2013). Dragonflies are known to be generalist predators of various insects (Corbet, 2004; Olberg et al., 2005; Kaunisto et al., 2017), which often house host-specific microbiota (Dillon & Dillon, 2004; Engel & Moran, 2013; Yun et al., 2014). Hence, a shift in the prey base due to geographical variation may directly or indirectly contribute to changes in the gut bacterial community of predatory dragonflies. We do acknowledge that among different environmental factors, seasonal variation can play a crucial role in gut bacterial community composition. However owing to our limited sampling effort we could not address the effect of seasonality in our present study, which remains to be tested in future studies.

Despite the major impacts of geographic location, concordant to our previous study (Nair & Agashe, 2016) we found that adults of each dragonfly host genus house a distinct gut bacterial community. What explains this partial host-specificity in the dragonfly gut microbiome? Our results suggest a potential role for phylogenetically conserved host level processes in shaping the gut community. For instance, both species from the genus Orthetrum shared a significant proportion of their gut community, whereas dragonflies from the genus Urothemis— the phylogenetically most distinct genus in our dataset (Ware, May & Kjer, 2007)— had very different gut bacteria. As is known for host taxonomy, host phylogeny could potentially structure insect gut microbiota through active or passive filters imposed by host morphology, physiology, development, immune function, social interactions or diet (Dillon & Dillon, 2004; Sullam et al., 2012; Colman, Toolson & Takacs-Vesbach, 2012; Jones, Sanchez & Fierer, 2013; Engel & Moran, 2013; Aksoy et al., 2014; Yun et al., 2014; Moran & Sloan, 2015); though this remains to be tested.

Broadly speaking, host-specific gut microbiota may reflect host specific diets and/or host specific selective filters (Colman, Toolson & Takacs-Vesbach, 2012; Engel & Moran, 2013). Unfortunately, information on dragonfly diet is scarce because of their rapid and unpredictable movements that make field observations difficult (Corbet, 2004). However, limited behavioral observations in natural and enclosed populations suggest that dragonflies are generalists (Fraser, 1933; Corbet, 2004; Stoks & Córdoba-Aguilar, 2012). A recent analysis of prey DNA from the faeces of three odonate species in Finland (including dragonflies and damselflies) found large dietary overlaps (Kaunisto et al. (2017), supporting the idea that odonates are generalists. In contrast, using a similar approach, we found that three common, sympatric dragonflies in India (O. sabina, O. pruinosum, and P. flavescens) consume very distinct insect communities. Our result is supported by prior behavioural observations (Fraser, 1933; Corbet, 2004) and our own observations of Orthetrum spp. preying on flies, and mosquitoes and P. flavescens consuming other dragonflies (personal observation of Orthetrum spp. and P. flavescens foraging, by Shantanu Joshi and Rittik Deb, year 2015–2016). These dietary differences were also strongly reflected in the diversity and richness of gut bacteria, suggesting a direct association between dietary and gut bacterial diversity. Our results suggest that each dragonfly species may have a unique dietary niche that acts as a passive filter modulating the entry of environmental microbiota into the gut.

However, we caution that our results indicate correlation rather than direct causation, and further work is necessary to assign causality.

Although this hypothesis requires further validation, we suggest that such dietary specialization— rather than strong host selection— is the primary driver of variation in dragonfly gut bacterial communities. Indeed, simulations using Sloan’s neutral assembly model (2006) revealed that bacterial communities were predominantly neutrally assembled with regard to host species and that assembly varied across geographic location and host species. Two additional lines of evidence support our conclusion that adult dragonfly gut bacterial communities are primarily structured via passive processes. First, the high taxonomic diversity of “selected” bacterial OTUs could suggest a lack of selection for a specific set of phylogenetically conserved bacterial functional traits (but see Louca et al. (2018)). Second, our FISH analysis also indicated a weak association with hosts. The rare observations of Wolbachia cells inside globular sacks in O. sabina deserve further attention as a possible special case of strong dragonfly-bacterial interactions. Nonetheless, our results indicate that although microbiota are transient, with likely limited impacts on the dragonfly host (comparable to observations in butterflies (Hammer et al., 2017; Phalnikar, Kunte & Agashe, 2018)), the microbiome may be largely shaped by host diet via its impact on microbial dispersal, nutrient availability and colonization .

Conclusions

Our analysis of the patterns of spatial and host-specific variation in the diet and gut bacterial communities of multiple wild-collected adult dragonflies highlights two key points. First, we suggest that environmental factors that may alter bacterial community stability should be given more importance when drawing general conclusions about host-microbe interactions. Second, while explaining variation in microbial community composition, it is important to explicitly consider neutral processes along with selection. The lack of significant host-microbiome relationships in dragonfly adults may arise from multiple reasons that remain to be tested. For instance, microbial mutualists may offer no benefit if dragonflies have endogenous digestive enzymes or acquire all essential nutrients from their prey. Alternatively, microbes may not be able to colonize the dragonfly gut because of the host’s variable and omnivorous diet, or for reasons of historical chance. We hope that our work encourages further analysis of variation in gut microbiomes of natural insect populations, as well as experimental tests of the role of neutral vs. selective processes in the assembly of host-associated microbial communities.

Data accessibility

All data are made available in public repositories. Sequencing data and metadata have been uploaded on the European Nucleotide Archive website (PRJEB32318, PRJEB32316, PRJEB32311, PRJEB32309 for gut bacterial data, PRJEB32308 for diet data). OTU tables are made available at Figshare (gut bacteria raw data file without any pruning: 10.6084/m9.figshare.9632084, diet raw data file without any pruning: 10.6084/m9.figshare.9632096).

Supplemental Information

Supplemental Information 1 Supplemental Figures and Tables

Click here for additional data file.

We thank the editor and the two anonymous reviewers for their detailed and insightful comments. We thank members of the Agashe lab for critically reading the manuscript. We thank Krushnamegh Kunte for contributing samples from Shendurney Wildlife Sanctuary; Agumbe Rainforest Research Station, Rohini Balakrishnan, and Sarita and Ramanuj Dasgupta for logistical support; Ashish Tiple, Krushnamegh Kunte, Kruttika Phalnikar, Manjunatha Reddy, Ronita Mukherjee, Sudhakar Gowda and Saira Guha for field assistance; Shantanu Joshi for help with identifying dragonflies; and Kruttika Phalnikar for help with QIIME analysis.

Additional Information and Declarations

Competing Interests

Author Contributions

Field Study Permissions

DNA Deposition

Data Availability

The authors declare there are no competing interests.

Rittik Deb conceived and designed the experiments, performed the experiments, analyzed the data, contributed reagents/materials/analysis tools, prepared figures and/or tables, authored or reviewed drafts of the paper, approved the final draft, sample collection and most of the experiments.

Ashwin Nair performed the experiments, contributed reagents/materials/analysis tools, authored or reviewed drafts of the paper, approved the final draft, sample collection.

Deepa Agashe conceived and designed the experiments, authored or reviewed drafts of the paper, approved the final draft, sample collection.

The following information was supplied relating to field study approvals (i.e., approving body and any reference numbers):

The Kerala Forest and Wildlife Department granted field permits for the Shendurney samples (permit no. WL 10-3781/2012 dated 18/12/2012, and GO (RT) No. 376/2012/F and WLD dated 26/07/2012). The rest of the samples were collected from outside of protected areas.

The following information was supplied regarding the deposition of DNA sequences:

Sequencing data and metadata is available at the European Nucleotide Archive. Gut bacterial data: PRJEB32318, PRJEB32316, PRJEB32311, PRJEB32309; Diet data: PRJEB32308.

The following information was supplied regarding data availability:

Deb, Rittik (2019): Open reference dragonfly gut bacteria data file without any pruning (raw format). figshare. Dataset. https://doi.org/10.6084/m9.figshare.9632084.v1.

Deb, Rittik (2019): Closed reference dragonfly prey item data file without any pruning (raw format). figshare. Dataset. https://doi.org/10.6084/m9.figshare.9632096.v1.

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
