# Peer review of "Host dietary specialization and neutral assembly shape gut bacterial communities of wild dragonflies"

_PeerJ, doi:10.7717/peerj.8058_

## Round 0.1 · original submission · Major Revisions

· Academic Editor

Major Revisions

Reviewer #1 points out some issues that are essential to address before this work is published in PeerJ. Please be sure to respond to all of their points, and the minor points raised by reviewer #2.

Reviewer 1 ·

Basic reporting

The authors claim that the data is deposited within the European Nucleotide Archive under the accessions: PRJEB32318, PRJEB32316, PRJEB32311, PRJEB32309, PRJEB32308. I am unable to access theses in order to determine if these data are compliant with MIMARKS standards (and am thus unable to determine if all associated meta data, analyzed within this manuscript, is provided for reproducibility of the work). I refer the authors to the following:

Yilmaz, Pelin, Renzo Kottmann, Dawn Field, Rob Knight, James R. Cole, Linda Amaral-Zettler, Jack A. Gilbert, et al. 2011. “Minimum Information about a Marker Gene Sequence (MIMARKS) and Minimum Information about Any (x) Sequence (MIxS) Specifications.” Nature Biotechnology 29 (5): 415–20.

Permits for field work appear to have been obtained from Shendurney Wildlife Sanctuary: permit no. WL 10-3781/2012 dated 18/12/2012, and GO (RT) No. 376/2012/F and WLD dated 26/07/2012). I have no way of confirming if these are appropriate permits. Additionally, it does not appear if permits were either required or obtained from the other sampling cites: Agumbe Rainforest Research Station, Bangalore, Nagpur, Bordubi.

Line 234: State here all of the specific R packages used. I realize some are listed later, but the specific commands / parameters used are quite vague on some places.

Line 236-237: What tool was used to perform rarefaction analyses? QIIME? An R package like Phyloseq or mctoolsr? It may behoove the authors to provide a simple ordered set of commands / flowchart detailing the analysis pipeline.

Lines 273-287: This seems like quite a robust approach! When describing the parameters, it’d be useful to show or reference the formula(s) in which these parameters are entered. Although the authors reference the R script from Burns et al, it is unclear which tables were used as input, i.e. what was considered the actual meta-community. Given the large amount of detail provided in other areas of this manuscript and within supplemental information, I am not sure why the authors chose not to also share their code / pseudo code / notebooks via github or other means too? I do not think I would be able to reproduce these steps.

Experimental design

Lines 110, 120, 127: To be clear separate individuals were used for each of these steps? That is, those used for FISH / qPCR (b) are different than those used for microbiome analysis (a)? If so, please explicitly state this.

Line 101 – 104: If I understand correctly, sampling was repeated from October through January at each location? How many times, please provide dates / time points. Otherwise, it is not clear how the assessment of community assembly can be performed without reference to a temporal component. Please clarify.

Line 120-123: I am a little confused by the statement referring to the 4-6 hours that the dragonflies were allowed to defecate and clear their guts. If that is all that is needed to remove most of the fecal material, then how can the later statement, that fecal material may represent a 3-5 day snapshot of diet be true?

Line 170: To be clear, the data set totals 45 dragonflies, or 45 dragonflies per species? IF the former, how many per species?

Lines 169-175 / 193-194: The authors assume that readers will understand why Hasora sp. is a “control”. This will not be clear to those not familiar with these molecular surveys. The authors should explicitly explain why this phytophagus taxon is being used as a control. They should also bring up the problems associated with potentially amplifying and sequencing host DNA, particularly when using the COX1 gene, and cite the appropriate literature. They also need to make note of what proportion of reads were from the host, versus food/prey items. I’d expect a box-whisker plot of how many reads are from the host compared to the total reads. The Kaunisto paper does reference the amount of host sequences obtained, which is quite substantial (71-84% are from host; Figure 1). I am concerned that the authors did not look into constructing a blocking primer or peptide nucleic acid clamp (PNA) to inhibit sequencing host DNA. These methods have been around for quite a while (especially for COX1) and the authors should state why this host-blocking approach was not implemented or not needed. Much of the recent advancement in limiting host amplification and sequencing to assess diet has been largely ignored by the authors within this manuscript. The authors should be referring, at the very least, to several of the following seminal papers in this field:
1) Vestheim, Hege, and Simon N. Jarman. 2008. “Blocking Primers to Enhance PCR Amplification of Rare Sequences in Mixed Samples – a Case Study on Prey DNA in Antarctic Krill Stomachs.” Frontiers in Zoology 5 (1): 12–11.
2) Deagle, Bruce E., Simon N. Jarman, Eric Coissac, François Pompanon, and Pierre Taberlet. 2014. “DNA Metabarcoding and the Cytochrome c Oxidase Subunit I Marker: Not a Perfect Match.” Biology Letters 10 (9): 20140562–20140562.
3) De Barba, M., C. Miquel, F. Boyer, C. Mercier, D. Rioux, E. Coissac, and P. Taberlet. 2014. “DNA Metabarcoding Multiplexing and Validation of Data Accuracy for Diet Assessment: Application to Omnivorous Diet.” Molecular Ecology Resources 14 (2): 306–23.
4) Kartzinel, Tyler R., Patricia A. Chen, Tyler C. Coverdale, David L. Erickson, W. John Kress, Maria L. Kuzmina, Daniel I. Rubenstein, Wei Wang, and Robert M. Pringle. 2015. “DNA Metabarcoding Illuminates Dietary Niche Partitioning by African Large Herbivores.” Proceedings of the National Academy of Sciences of the United States of America 112 (26): 8019–24.
5) Lundberg, Derek S., Scott Yourstone, Piotr Mieczkowski, Corbin D. Jones, and Jeffery L. Dangl. 2013. “Practical Innovations for High-Throughput Amplicon Sequencing.” Nature Methods 10 (10): 999–1002.
6) Piñol, J., V. San Andrés, E. L. Clare, G. Mir, and W. O. C. Symondson. 2014. “A Pragmatic Approach to the Analysis of Diets of Generalist Predators: The Use of next-Generation Sequencing with No Blocking Probes.” Molecular Ecology Resources. https://doi.org/10.1111/1755-0998.12156.
7) Brandon-Mong, G-J, H-M Gan, K-W Sing, P-S Lee, P-E Lim, and J-J Wilson. 2015. “DNA Metabarcoding of Insects and Allies: An Evaluation of Primers and Pipelines.” Bulletin of Entomological Research 105 (6): 717–27.
Note, I am not criticizing the use of COX1 or lack of blocking primers/probes, only that the authors should bring up known limitations using these approaches, then discuss whether or not they think these limitations may or may not affect the authors overall interpretations. I suspect likely not.

Lines 205-207: I think the authors are confusing de novo, open-reference, and closed-reference OTU-picking. It appears they only made use of the open and closed reference approaches. Although de novo is a component of open-reference, the wording of this sentence makes the reader assume three methods are being compared. I am generally fine with any of these approaches, however, there is strong potential for the count inflation of some OTUs using these approaches. Or, more to the point, the dropping of reads that to not “hit” the database, not necessarily because they are not within the database, but the stopping criteria is met before an actual “hit” is found. I hope the authors adjusted the `max_accepts` and `max_rejects` flags of the open and close reference approaches by setting them to larger values or setting them both to `0` (exhaustive search). Though this is more of an issue for the closed-reference approach. By this, I mean that the authors may not be aware of how usearch / uclust actually works. I highly recommend reading this online blog post by Scott Olesen entitled “What you think usearch does might not be what it does.”
http://scottolesen.com/post/what-you-think-usearch-does-might-not-be-what-it-does/
Again, I am not dissuading the use of closed reference approaches, but they authors should be aware and point out the parameter settings used, as well as the caveats associated with these settings. There is literature on the subject, but I simply felt the blog post is a more “fun” way to learn about this. Finally, the authors need to make sure they cite all of the tools used that are contained within QIIME and/or the R packages. For example, the authors do not cite uclust or the open /closed reference approach:
1) Edgar, Robert C. 2010. “Search and Clustering Orders of Magnitude Faster than BLAST.” Bioinformatics 26 (19): 2460–61.
2) He, Yan, J. Gregory Caporaso, Xiao-Tao Jiang, Hua-Fang Sheng, Susan M. Huse, Jai Ram Rideout, Robert C. Edgar, et al. 2015. “Stability of Operational Taxonomic Units: An Important but Neglected Property for Analyzing Microbial Diversity.” Microbiome 3 (1): 20.

Lines 214-217: The easy-to-access / download files from the Barcode of life is not very complete. In fact there is much more information that can be accessed via the R package “bold”, where this data should be downloaded (if not done so already). Even better, I highly recommend that the authors use the reference database constructed by Teresita Porter:
1) https://github.com/terrimporter/CO1Classifier
2) Porter, Teresita M., and Mehrdad Hajibabaei. 2018. “Automated High Throughput Animal CO1 Metabarcode Classification.” Scientific Reports 8 (1): 4226.
3) Porter, Teresita M., and Mehrdad Hajibabaei. 2018. “Over 2.5 Million COI Sequences in GenBank and Growing.” Edited by Wolfgang Arthofer. PloS One 13 (9): e0200177.

Line 410: Clarify how many individuals per dragonfly species were sequenced. This sentence is ambiguous, do you mean 5 individual hosts or 5 individual host species? If only 5 hosts, then I am concerned that this is not a large enough sample size to make robust inferences. Probably a good idea to perform a power analyses using micropower to determine effect size in future studies.

Validity of the findings

Line 217-221: I agree with the authors that removal of Odonate sequences may hide any dietary / ecological insight related to conspecific predation. I thank the authors for making this really good point! Again, although clear to me, many others may not understand the issue of obtaining a large amount of sequence from the host. This must be explicitly stated in the manuscript. However, it’d be nice to simply know the percentage of general Odonate sequences obtained that do not specifically classify to the host sample. They could conjecture that this “might be” a signal of conspecific predation. Actually, the authors could use this to address my earlier comment about blocking primers, in part. That is, blocking primers (i.e. Vestheim) will likely block the conspecific predation signal. But if they are not concerned with this issue (or it is too difficult to discern given the particular amplicon region chosen to access diet), then a blocking primer should be considered / discussed. However, if they are interested in blocking host DNA, then this would only argue for the use of PNA, which can be tailored to be species-specific as they are far less “leaky” (i.e. blocking other off-targets) than blocking primers.

Line 609-611: What dates were these sites sampled? I am concerned that temporal variability in diet may conflate “partitioning out” the effects of the role of geography and diet. Can more information be provided on the prevalence of diet items within each region? Do they fit with the diet items found via COX1? Temporal variation in food item availability has a large impact on feeding behavior / diet.

Lines 58-61: This is over generalized. Even generalists will focus on a “favorite” or preferred food item as the season change. See the literature related to pigs, for example. I’d assume the same is potentially true for some insects too. I think the authors should cite other references, that specifically discuss these topics. The references used here to not appear to make any claims, nor do the studies seem to be explicitly designed to address ecological community assembly processes of the gut microbiota, e.g. stochastic assembly. Furthermore, any stochasticity observed in these studies is conflated, in large part, by the fact that the (greatly summarized by the cited Yun paper) “…insect gut as a bacterial habitat shows different morphologies, generally depending on the insect order; furthermore, the gut morphology is sharply changed by metamorphosis, according to the life cycle of the insect. Additionally, the oxygen availability can be influenced by gut shape, metabolism of colonizing bacteria, and partial pressure of oxygen from the outside environment, and the pH, from acidic to extremely alkaline conditions, is determined by different gut compartments in diverse insect individuals. These diverse gut conditions may cause the variation in host-specific gut microbiota in insects.”

Lines 80-82: I think the authors should more clearly define what they mean by “generalist predators” as this term can be ambiguous, and often considered a redundant term. Do the authors simply mean generalist in the sense of, eating many different species? For example, a predator can feed on many different species (generalist in reference to species), but limited by the size of the prey species, or even the nutrient composition of the prey species. So, in this sense, the predators can be considered specialists, e.g. eating only prey items they can handle within a certain size range. Also, they may be generalists in what they consume throughout the year but, they prey upon items that are abundantly available in a given season (i.e. seasonal specificity), or the nutrients the prey items provide. Predators typically have more plasticity compared to their herbivorous counterparts. Again, even generalists will focus on a “favorite” or preferred food item when present. I refer the authors reader to:
1) Unsicker SB, Oswald A, Köhler G, Weisser WW. Complementarity effects through dietary mixing enhance the performance of a generalist insect herbivore. Oecologia. 2008;156(2):313–324. doi:10.1007/s00442-008-0973-6
2) Jaworski CC, Bompard A, Genies L, Amiens-Desneux E, Desneux N (2013) Preference and Prey Switching in a Generalist Predator Attacking Local and Invasive Alien Pests. PLOS ONE 8(12): e82231. https://doi.org/10.1371/journal.pone.0082231

Lines 297-300: The authors are making an assumption that functional redundancy only exists among phylogenetically related taxa. This is not always the case. There is a decent amount of literature on the topic:
1) Louca, Stilianos, Martin F. Polz, Florent Mazel, Michaeline B. N. Albright, Julie A. Huber, Mary I. O’Connor, Martin Ackermann, et al. 2018. “Function and Functional Redundancy in Microbial Systems.” Nature Ecology & Evolution 2 (6): 936–43.
2) Koppel, Nitzan, Vayu Maini Rekdal, and Emily P. Balskus. 2017. “Chemical Transformation of Xenobiotics by the Human Gut Microbiota.” Science 356 (6344). https://doi.org/10.1126/science.aag2770.

Lines 378-381: Given the range of sequencing depth for the samples used as presented on line 205 (8k-900k) and 214 (29k-115k), even in non-rarefaction approaches the read depth shouldn’t not deviate excessively. That is: “Alternate normalization measures are potentially vulnerable to artifacts due to library size.” (Ref 3 below). There are quite a lot of issues with non-rarefaction approaches if not performed or investigated properly. One of the factors, I suggest the authors read the following for more details:
1) Weiss, Sophie J., Zhenjiang Xu, Amnon Amir, Shyamal Peddada, Kyle Bittinger, Antonio Gonzalez, Catherine Lozupone, et al. 2015. “Effects of Library Size Variance, Sparsity, and Compositionality on the Analysis of Microbiome Data.” e1408. PeerJ PrePrints. https://doi.org/10.7287/peerj.preprints.1157v1.
2) Weiss, Sophie, Will Van Treuren, Catherine Lozupone, Karoline Faust, Jonathan Friedman, Ye Deng, Li Charlie Xia, et al. 2016. “Correlation Detection Strategies in Microbial Data Sets Vary Widely in Sensitivity and Precision.” The ISME Journal 10 (7): 1669–81.
3) Weiss, Sophie, Zhenjiang Zech Xu, Shyamal Peddada, Amnon Amir, Kyle Bittinger, Antonio Gonzalez, Catherine Lozupone, et al. 2017. “Normalization and Microbial Differential Abundance Strategies Depend upon Data Characteristics.” Microbiome 5 (1): 59.
Additionally, the approach referenced by the authors may not perform well when considering that their data may be zero-inflated. Even when using variance / log-like transformations the signal of the original sequencing depth can still be present within the data and may affect downstream analyses. One great example of this is referred to below in Nieves-Ramirex et al. (be sure to also check the accompanied supplementary information / data). In fact, the Nieves-Ramírez paper used the approaches outlined in the references above. Thus, any analyses based on non-rarefied data need to be vetted. Which can also be an issue for differential abundance analyses (i.e. potential false positives). Again, these issues are outlined in the references above. Why was DESeq2 used over the plethora of other differential abundance tools? I’d recommend making use of DATest (Ref 2 below), to guide the most appropriate differential abundance based approach given your data (e.g. does your data fit a negative binomial approach, zero-inflated,… etc?).
1) Nieves-Ramírez, M. E., O. Partida-Rodríguez, I. Laforest-Lapointe, L. A. Reynolds, E. M. Brown, A. Valdez-Salazar, P. Morán-Silva, et al. 2018. “Asymptomatic Intestinal Colonization with Protist Blastocystis Is Strongly Associated with Distinct Microbiome Ecological Patterns.” mSystems 3 (3).
2) Russel, Jakob, Jonathan Thorsen, Asker D. Brejnrod, Hans Bisgaard, Søren J. Sørensen, and Mette Burmolle. 2018. “DAtest: A Framework for Choosing Differential Abundance or Expression Method.” bioRxiv, 241802.

Additional comments

The manuscript entitled “Host dietary specialization and neutral assembly shape gut bacterial communities of wild dragonflies” by Deb et al., attempts to determine the factors that may affect the composition of microbiota among several species of dragonflies. These factors included, host identity, geographic location, and diet. The authors made use of a variety of methods to parse out these effects, including the use neutral models of community assembly.

I have some concerns with the overall interpretations of the results given the very limited sample size of the study. The fact that this study was not planned as a longitudinal study is problematic for many of the authors interpretations. The authors do own up to this towards the end of this manuscript. Was the data from Nair & Agashe 2016 used for this study too? The authors used the approach of Burns et al. 2016 for the assessment of community assembly. However, unlike Burns et al 2016 (and their own prior work) I was unable to follow how many time points where used for this study. Being that the authors focus much of their work on community assembly, I am surprised more clear details were not provided on how many time points they sampled. I plot similar to Figure 2 of Burns et al 2016 would be ideal.

The authors clearly did their due diligence with the FISH probes, alongside the assessment of microbial community and diet analyses… appears well thought out. The amount of work provided by the authors is quite substantial! I hope that, my somewhat long review, will help refine this already compelling piece of work.

Line 31-33: Clarify. Is this referring to the gut microbiome in general, or only those microbes that are ‘attached’ to the gut lining?

Line 34: The authors should state their overall results with explicit reference to their study system, and not generalize. Rephrase the sentence: “Our results contradict the expectation that host-imposed selection shapes gut microbiota, and highlight the importance of joint analyses of diet and gut microbiota of natural host populations.” To: “Our results contradict the more general expectation that host-imposed selection shapes gut microbiota of dragonflies, and highlight the importance of joint analyses of diet and gut microbiota of natural host populations.”

Lines 56-58: This is a very general statement. Provide at least one or two in-text examples of how dietary specialists continually promote these things.

Lines 183-187: When listing the primer sequences used, be courteous to your readers, please insert spaces or hyphens between the components of the primers, i.e. adapters, actual primer sequence, etc…

Lines 224-231: I really like this approach!

Lines 247-248: The authors bring up non-rarefaction approaches later in the manuscript, why not use such an approach for beta-diversity analyses too, such as Aitchison PCA:
1) Martino, Cameron, James T. Morton, Clarisse A. Marotz, Luke R. Thompson, Anupriya Tripathi, Rob Knight, and Karsten Zengler. 2019. “A Novel Sparse Compositional Technique Reveals Microbial Perturbations.” mSystems 4 (1): e00016–00019.

Also, the authors should cite the original developers of the Canonical Analysis of Principal Coordinates based on Discriminant Analysis method:
1) Anderson, M. J., and T. J. Willis. 2003. “Canonical Analysis of Principal Coordinates: A Useful Method of Constrained Ordination for Ecology.” Ecology. https://esajournals.onlinelibrary.wiley.com/doi/abs/10.1890/0012-9658(2003)084%5B0511:CAOPCA%5D2.0.CO%3B2.

Line 256-264: How were these performed? Cite / refer to the tools and parameters used at each step. Again, QIIME, R packages? Other scripts?

Line 262-264: Unclear to me why this combination of analyses are being used. Specifically, why perform a post-hoc test after Kruskal-Wallis?

Line 408: “OUT” should be “OTU”.

Lines 513-517: Maybe, though the sample size is quite low. Not entirely convinced, as the proper approach for determining community assembly is to perform a longitudinal survey. This feels more like a cross-sectional study, with location (pooled data) and sampling time conflating any robust findings. If I am mistaken, please clarify.

Lines 553-560: I think the authors are assuming that microbial host specialists (if there are any) will be bound to the gut lining. This may not always be true. Bacteria are motile organisms, and like trout swimming upstream, can swim in the opposite direction of fecal passage. Thus, it could be possible that microbiota can be host specific, yet reside within or on the surface of fecal material as new material is formed. With any metabolomic cross talk still occurring. I would assume that bacteria attached to the gut lining depend upon what is in the gut. So, very much like the monsoon and dry seasons of desert landscapes, the same can be said here. That is, I would expect the abundance and type of microbial species remaining on the lining of the gut to become more depauperate. Thus, we are looking at the microbiota of the gut lining that can deal with “droughts”, not necessarily those that may be able to remain. It’d be great if the authors can speak to this. Especially, considering the strong assumption that host specificity is tied to being able to “attach” to the gut lining. To be sure, this is a reasonable assumption, but not sure how well vetted this assumption really is…

Lines 570-584: Need to refer to the figures / tables throughout this section.

Lines 583-586: This is the major weakness of the study. Although acknowledged by the authors, they maintain emphatic statements throughout the manuscript, e.g. “This result contradicts the hypothesis that dragonflies impose strong positive selection favoring a specific, shared set of functionally important bacteria”. Which is a bit disingenuous. You cannot have it both ways. I suggest that the statements be made less emphatic.

Lines 595-600: Why not test some of these potential associations in part? There are a variety of approaches. Blombergs K, UniFrac, Phylofactorization, etc… Some other examples: can map microbial community alpha diversity or microbial community states (i.e. microbiotyping) with host phylogeny.

Line 621-623: This is great, and I largely agree with this! I feel that this point should be made more clearly as potential outcome. This partly ties in with my comments about microbiota that may live in/on the fecal material rather than be attached to gut lining. To be honest, this point (diet specialization is a strong driver of gut bacterial communities) is a bit lost throughout the entirety of the manuscript. Perhaps this lack of clarity is due to too much reference to neutral assembly. This leads the reader, including this one, to assume that the point being made has to do with lack of differences in microbial communities among the dragonflies across the regions. This appears to not be the case. The microbiota are different, but not due to host species type, or geography, but due to diet.

Line 632-633: This statement, on the surface, seems to contradict what was referred to in my previous comment. I’d recommend being explicit by stating / restating that although microbiota are transient, with likely limited impact on the host, they are indeed driven by diet (i.e. nutrients available to the microbiota). Alternatively, could there be alternate stable states of gut microbiota?

Line 639-641: True, but selection for what. Remember selection is contextual. Selection can be occurring, but is a culmination of, season, diet, host, etc… The microbial communities are different because of different diet….

Line 646-648: It sure will. This is exciting research!

Fire S1 – should place within the figure legend the number of samples.

Supporting file line 122: should be “closed reference”

Reviewer 2 ·

Basic reporting

I found this paper easy to read, with clear writing and appropriate literature citations.

Experimental design

Experimental design is impressive, even though the spatial and temporal resolution of the study are somewhat limited for reasons of practicality.

I also commend the authors for presenting their findings even when their FISH experiment did not show strong results.

Validity of the findings

The authors do a good job of explaining their findings in context of the limitations of their study. The discussion of spatial and prey patterns in their gut microbiome data is appropriately nuanced, and the authors set up themselves or others nicely for future studies looking at environmental conditions (temperature, vegetation, etc) that may be underlying the large effect of sample site reported here.

Additional comments

I only have minor comments on this manuscript, as I do not have any major concerns. I will note that I am not an expert in the field of entomology and I do not have much experience with FISH, so I will not comment on those areas of the manuscript.

L48: What does "through conspecifics" mean in this context? Does it refer to interactions with conspecific dragonflies? Please clarify.

L76: This sentence seems to imply universality in the environmental/host/geographic factors structuring andimal gut bacterial communities. However, it is more likely that assembly mechanisms are very different depending on what species of animal is being considered.

L97: It is unclear what the indiabiodiversity.org citation is for. Please clarify within the parenthetical.

L199: Please specify what version of QIIME software was used, and whether there was a specific protocol or pipeline that was used. Reference to a paper from which the authors drew bioinformatic inspiration would also be fine.

L202: Citation for Fast-QC?

L245: ADONIS is in the R package "vegan"

L246: Please explain what "near-zero variance" means and why it was desirable to remove it from the data.

L293: I agree that within the authors' experimental design, OTUs can occur at a higher frequency within host than expected given the "metacommunity" of OTUs found across all hosts (positive selection). To be clear, the actual metacommunity is the pool of OTUs that are reasonable to assume could be found inside dragonfly guts, and this metacommunity is only estimated by the pool of all OTUs found in the study (L276). However, I do not agree that negative selection can be inferred by this study. An OTU cannot be both "abundant within the metacommunity" and "occur in only a few individuals", because for that OTU to be abundant in the metacommunity it would have to occur within many individuals. This comment is not a major issue for me, because the authors do not discuss negative selection in any significant manner.

L376: Consider using "collectors' curves" instead of "rarefaction analysis" since this is the language used earlier in methods (and I think it's a clearer term as well)

L407: clarify what "environmental factors" are or could be, since the term is vague.

L446: it would be helpful to describe beta-diversity in this context as within-group (within-site) variability.

L502: Unclear how binomial model indicates that dispersal (instead of environmental selection) contributes to beta diversity

L551: citation for host selection stabilizing communities in the face of env variation?

L558: Consider rephrasing "predominantly neutrally assembled" with "neutrally assembled with regard to host species", since community assembly is certainly not neutral with regard to sample site and also could be (and is likely) under selection by unknown environmental factors. Same on L624.

---

## Round 0.2 · Minor Revisions

· Academic Editor

Minor Revisions

Both reviewers have approved publication of this manuscript. After a final review that I performed, I remain concerned about your decision to not rarify your data before alpha and beta diversity analysis (I am looking at lines 417-418 - please correct me if I am misinterpreting) - this was initially brought up by Reviewer 1. It is well known that sampling depth can impact alpha and beta diversity. Quoting from the McMurdie paper that you reference:

"Some of the justification for the rarefying procedure has originated from exploratory sample-wise comparisons of microbiomes for which it was observed that a larger library size also results in additional observations of rare species, leading to a library size dependent increase in estimates of both alpha- and beta-diversity, especially UniFrac. It should be emphasized that this represents a failure of the implementation of these methods to properly account for rare species and not evidence that diversity depends on library size. Rarefying is far from the optimal method for addressing rare species, even when analysis is restricted solely to sample-wise comparisons. As we demonstrate here, it is more data-efficient to model the noise and address extra species using statistical normalization methods based on variance stabilization and robustification/filtering."

The authors are not saying that it's acceptable to run alpha and beta diversity metrics without accounting for differing library size, but rather that rarifying is a wasteful way of account for differing library size.

I recommend that you revisit the decision to not rarify before alpha and beta diversity, or that you apply one of the alternative normalization procedures highlighted in the McMurdie paper. I'm ok with the unrarified data remaining in the paper, as long as corresponding plots for Figures 2 and 3 with rarified or otherwise normalized data are also present (either unrarified or rarefied/normalized could be supplemental). I would like the see your findings confirmed across these two parallel analyses.

After this is addressed, I can make a rapid editorial decision on this manuscript without sending the paper out for re-review.

Reviewer 1 ·

Basic reporting

English writing and content is clear.
The authors updated literature references as requested.
Raw data is easily accessible. Kudos to the authors for providing, not only links to the raw data, but their OTU tables too.
Results and author conclusions appear to be commensurate.

Experimental design

The authors went above and beyond clarifying their approach.
The authors were quite thorough, given the study system, in their approach to combine field sampling, FISH, diet analysis, and a 16S rRNA microbiota survey.
Research does provide a great foundation to start compelling research with insect host-gut relationships.

Validity of the findings

All raw data appear to have been provided. I assume the figshare links to the OTU tables will be released upon acceptance of the manuscript (I was unable to access them).
Conclusions, and associated caveats, were well described and highlighted. I greatly appreciate the authors due diligence in this regard.

Additional comments

I would like to thank the authors for their diligence and thorough response to the reviewers concerns and comments. I am more than satisfied with their considerate responses. I think the revised work is more clear and quite compelling. The willingness of the authors to redo some of the analysis, just to confirm their findings was quite refreshing. I’ve nothing more to suggest / add to the manuscript. I wish the authors best of luck in their research.

Reviewer 2 ·

Basic reporting

No issues here.

Experimental design

Still looks good to me. Authors adequately addressed my requests for clarification where appropriate.

Validity of the findings

Authors present their findings in a very straightforward and honest way, and refrain from over-interpreting their results. No problems here.

Additional comments

All my issues with the paper have been addressed.

---

## Round 0.3 · accepted · Accept

· Academic Editor

Accept

Thank you for addressing my final comments on the rarefaction analysis.